# The dehydration carousel of stratospheric water vapor in the Asian Summer Monsoon Anticyclone

Paul Konopka[1], Christian Rolf[1], Marc von Hobe[1], Sergey M. Khaykin[3], Benjamin Clouser[4], Elisabeth Moyer[4], Fabrizio Ravegnani[5], Francesco D'Amato[6], Silvia Viciani[6], Nicole Spelten[1], Armin Afchine[1], Martina Krämer[1], Fred Stroh[1], and Felix Ploeger[1,2]

[1]Forschungszentrum Jülich, IEK-7, Jülich, Germany
[2]Institute for Atmospheric and Environmental Research, University of Wuppertal, Wuppertal, Germany
[3]Laboratoire Atmospheres, Observations Spatiales (LATMOS), CNRS/INSU, Sorbonne Universite, Guyancourt, France
[4]Department of the Geophysical Sciences, University of Chicago, Chicago, IL, USA
[5]National Research Council - Institute for Atmospheric Sciences and Climate (ISAC-CNR), 40129 Bologna, Italy
[6]National Institute of Optics, CNR-INO, Via Madonna del Piano 10, Sesto Fiorentino, Florence, Italy

**Correspondence:** Paul Konopka (p.konopka@fz-juelich.de)

**Abstract.** During the StratoClim Geophysica campaign, air with total water mixing ratios up to 200 ppmv and ozone up to 250 ppbv was observed within the Asian summer monsoon anticyclone up to 1.7 km above the local cold point tropopause (CPT). To investigate the temporal evolution of enhanced water vapor being transported into the stratosphere, we conduct forward trajectory simulations using both a microphysical and an idealized freeze-drying model. The models are initialized at the mea-
surement locations and the evolution of water vapor and ice is compared with satellite observations of MLS and CALIPSO. Our results show that these extremely high water vapor values observed above the CPT are very likely to undergo significant further freeze-drying due to experiencing extremely cold temperatures while circulating in the anticyclonic "dehydration carousel". We also use the Lagrangian dry point (LDP) of the merged back-and-forward trajectories to reconstruct the water vapor fields. The results show that the extremely high water vapor mixed with the stratospheric air has a negligible impact on the overall
water vapor budget. The LDP mixing ratios are a better proxy for the large-scale water vapor distributions in the stratosphere during this period.

## 1 Introduction

Stratospheric water vapor (SWV) is a potent greenhouse gas with a significant radiative forcing (Forster and Shine, 1999; Solomon et al., 2010; Dessler et al., 2013). In the tropical lower stratosphere, SWV values are determined primarily by the
freeze drying of moist tropospheric air entering the stratosphere at the cold point tropopause (CPT) (Brewer, 1949; Randel and Park, 2019; Smith et al., 2021). The extent of ice injected into the stratosphere through deep, overshooting convection remains uncertain (Randel et al., 2012; Avery et al., 2017; Ueyama et al., 2020; Jensen et al., 2020; Ueyama et al., 2023). Specifically, in the context of the Asian summer monsoon (ASM) region, recent in-situ measurements suggest moistening due to convective activity above the local tropopause (Khaykin et al., 2022), while satellite observations indicate a broader-scale drying effect
caused by convection (Randel et al., 2015).

Lagrangian studies commonly reconstruct SWV by tracking the minimum saturation mixing ratio of air at the Lagrangian dry point (LDP) (Fueglistaler and Haynes, 2005; Liu et al., 2010; Schoeberl and Dessler, 2011; Smith et al., 2021). We combine airborne in situ measurements during the StratoClim campaign in Nepal (Lauther et al., 2021) with satellite observations of MLS (Livesey et al., 2020) and CALIPSO (Vaughan et al., 2009) to investigate the representativeness of the moist air masses encountered above the CPT in order to reconstruct the large-scale SWV distribution. Using a microphysical model along forward trajectories, we address two main questions: (i) How does the water vapor content of these air masses change during their ascent into the stratosphere? (ii) How representative are these air masses in order to reconstruct the large-scale moisture budget of the lower stratosphere? Finally, we discuss the performance of LDP-based SWV reconstructions for these examples of moistening above the local CPT.

## 2  In situ data analysis: Cold Point Tropopause (CPT) and Lagrangian Dry Point (LPT) perspectives

In this study, we utilize in situ data collected during all local Geophysica flights over Nepal in 2017, combined with merged back-and-forward trajectories driven by the ERA5 reanalysis (Hersbach et al. (2020), Appendix A), to evaluate the influence of these air masses on SWV values. In Figure 1a, two thick black lines define data points that are "sufficiently moist" ($H_2O > 7$ ppm) and "sufficiently deep in the stratosphere" ($O_3 > 100$ ppb). Our data set is reduced to 2315 data points by only considering those observed above the local CPT defined as the temperature minimum in the ERA5 temperature profiles interpolated to the Geophysica flight track. We denote the data with recent convective influence as type A (observed on Aug. 10) and with aged convective influence as type B (observed on Jul. 29). The data observed on Aug. 8 (less than 5%) contain mixed properties of types A and B and are labeled as type M (see Table 1).

Figure 1b shows CO values for all three data sets as a function of vertical distance to the CPT. CO decreases with altitude in slowly ascending air within the ASM anticyclone due to its chemical lifetime of a few months (Minschwaner et al., 2010; von Hobe et al., 2021). This decrease can be seen for type B and M data, extending up to 1.7 km above the tropopause. Type A data are closer to the CPT and show fresh convection signatures, with a spread of CO values between 30 and 95 ppb, large spread of HDO/$H_2O$ ratios (Figure A1c), and positive ice-CO correlations (Figure A1d). In Figure 1c, the distribution of the LDP ages derived from merged back-and-forward ERA5 trajectories within $\pm 60$ days is presented. Negative (positive) age denotes past (future) occurrence relative to the observation time. Consistent with our interpretation of recent and aged convection signatures, LDP ages range between -15 and -35 days for type B. For type A, more than 75% of the LDP encounters are still expected to occur along the forward trajectories, with LDP ages of 0-3 days, despite these air parcels have been sampled above the local CPT.

## 3  Dehydration scenarios along the forward trajectories

We use forward trajectories starting from the locations of the observed values of water vapor and ice for all types of air, A and B. Along these trajectories, we apply a state-of-the-art microphysical box model, CLaMS-Ice (Appendix A), as well as a

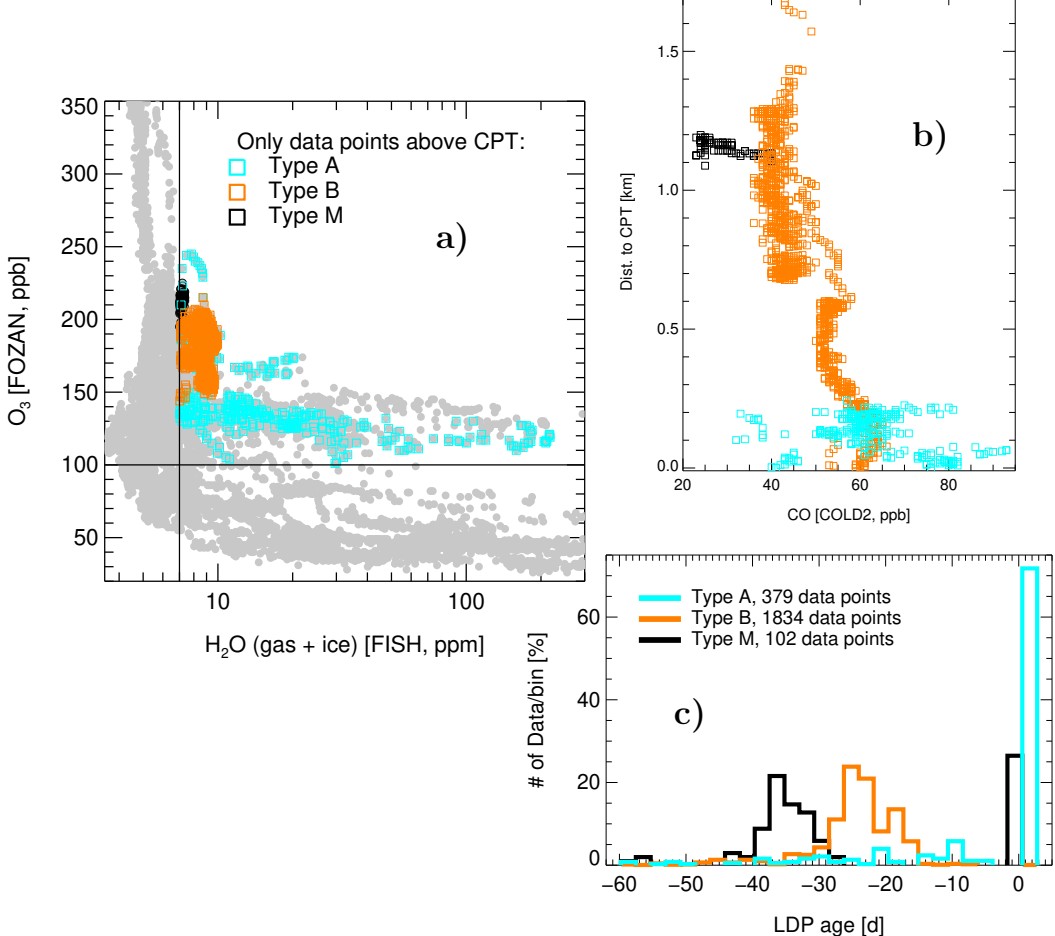

**Figure 1.** (a) H$_2$O-O$_3$ correlations for all local StratoClim flights (gray) with color-coded data points above the local cold point ERA5-tropopause (CPT) exhibiting total H$_2$O mixing ratios (gas + ice) greater than 7 ppm and O$_3$ mixing ratios greater than 100 ppm, divided into three groups: type A, type B, and type M (see text for further details). (b) Vertical distance to the local CPT as a function of CO for all three data types. (c) Normalized frequency distributions of Lagrangian dry point (LDP) ages derived from merged back-and-forward ERA5 trajectories within ±60 days. Negative (positive) age indicates that the LDP was found in the past (future) relative to the observation time (see text and Table 1 for details).

simple freeze drying model (FDM) that instantaneously removes excess water vapor along the forward trajectories when the air becomes supersaturated with respect to ice. Figure 2a shows two exemplary forward trajectories (type A and B) that slowly ascend within the ASM anticyclone above the level of zero radiative heating (Vogel et al., 2019) with a rotation period of about 10 days (Legras and Bucci, 2020). All trajectories of type A and more than 85% of type B stay within the tropical band extending northward during the boreal summer up to ∼40N. Only ∼15% of the type B trajectories descend into the lowermost

|  | Type A (fresh convection) | Type B (aged convection) | Type M (mixed) |
|---|---|---|---|
| Flight dates (2017) | Aug. 10 | Jul. 29 | Aug. 8 |
| Number of data | 379 | 1834 | 102 |
| Distance to CPT | 0–0.25 km | 0.0–1.7 km | 1.1-1.3 km |
| 75% of LDP ages | 0 to 3 days | -35 to –15 days | bimodal |
| $H_2O$ (gas) | 3.4-6.1 ppm | 7.0-10.2 ppm | 6.9-7.3 ppm |
| $H_2O$ (ice) | values up to 200 ppm | no ice observed | values up to 0.16 ppm |
| CO | 32-93 ppb, strong spread | 36-66 ppb, moderate variab. | 23-40 ppb, weaker variab. |
| CO-ice correlation | positive and significant | no correlation | no correlation |
| $HDO/H_2O$, delta D | -700 to -300‰, strong spread | around -400‰, weaker variab. | around -480‰, moderate variab. |

**Table 1.** Differences between moist signatures of air with fresh (type A) and aged convection (type B) observed above the CPT. Type M shows mixed properties of type A (∼30%) and B (∼70%) with a bi-modal distribution of the LDP ages (Figure 1c). Thus, while the LDPs of the type B air masses lie clearly in the past, type A may experience the strongest dehydration also in the future. For more experimental details see Figure A1.

stratosphere (LMS) northward of ∼40N (Figure B1a) after being detached from the anticyclone. During their spiraling ascent, almost all trajectories repeatedly pass through regions with low temperatures, well below 195 K, mainly on the south-eastern, southern, and south-western flank of the anticyclone, where water condensation and ice formation can occur. CLaMS-Ice and 

a simple freeze drying model (FDM) were used to simulate the trajectories and investigate dehydration scenarios. Figure 2c compares the models' results for one trajectory of type A with ice and water vapor observations from CALIPSO and MLS, respectively. CLaMS-Ice reproduces the CALIPSO ice observations fairly well, while FDM performs better in terms of water vapor comparison with MLS.

  We extend the analysis to all type A and B forward trajectories initialized with in-situ observations of $H_2O$ vapor and ice.

Figure 3 displays the time-dependent frequency distribution at selected times for type A (left) and type B (right). Relative to the initial distribution, the distributions derived from CLaMS-Ice and FDM evolve over time by moving to significantly lower values of total $H_2O$. The contribution of the dehydration driven by the parameterized gravity waves (GW) temperature fluctuations is very weak, as estimated from the small difference between the type B CLaMS-Ice distributions after 5 versus after 40 days (this difference vanishes if this parameterization is switched off). The 40-day distributions also show the impact

of enhanced ice nucleation by using CLaMS-Ice-IN.

  Despite the GW parameterization in CLaMS-Ice, the results of FDM are always drier than the results of CLaMS-Ice. This can be attributed to the different treatment of ice within the two models. In FDM, ice is removed instantaneously while in CLaMS-Ice ice removal occurs through the interplay of condensation, evaporation, and sedimentation (see also Figure 2c). A massive dehydration can be diagnosed for type A, affecting all air parcels, with mean/maximum values after 40 days of

5.0/11.3 ppmv (CLaMS-Ice) and 3.3/4.1 (FDM). The degree of dehydration for type B is weaker, as there are less than 1 ppm

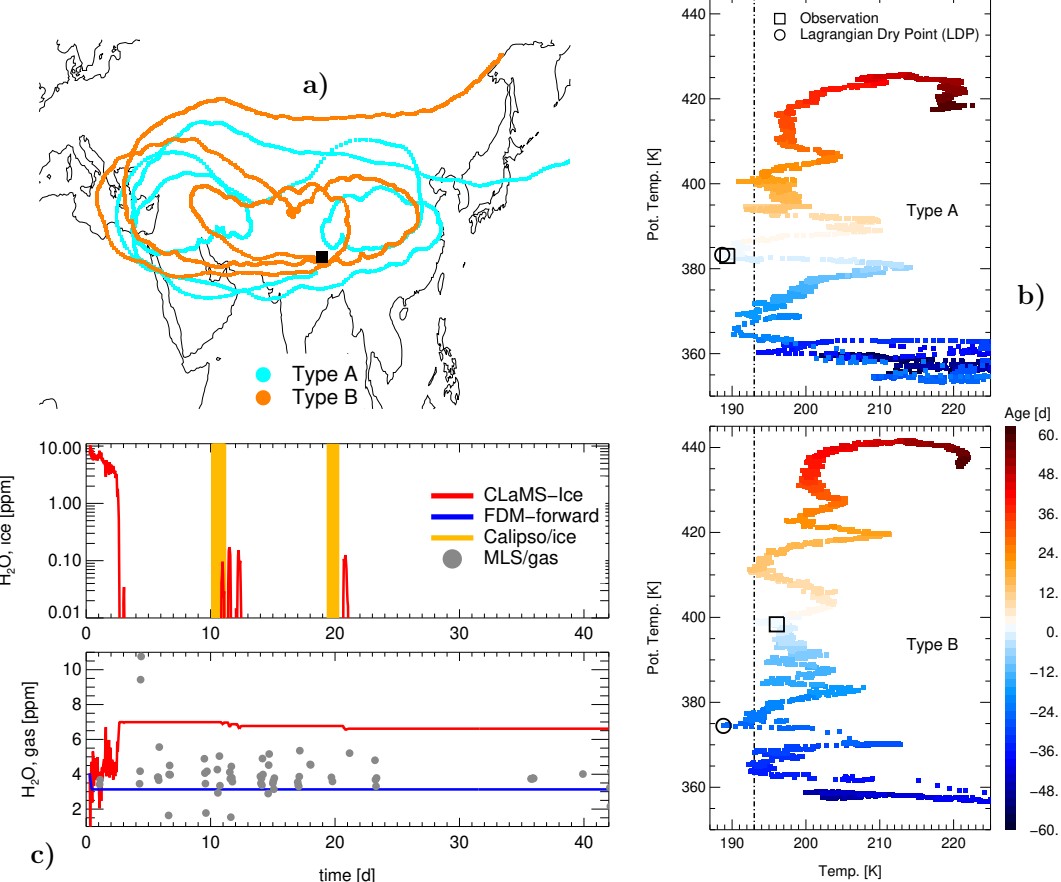

**Figure 2.** Box model studies comparing ice formation using CLaMS-Ice and a simple freeze drying model (FDM). (a) Horizontal view of two representative, forward trajectories classified as type A and B. (b) Temperature evolution along the merged back-and-forward trajectories, color-coded by trajectory age, with marked Lagrangian dry points (LDPs) diagnosed 1 day in the future for the type A trajectory and 12 days in the past for the type B trajectory. These two selected trajectories A and B start form the observed values of $H_2O$ (gas 7.13/9.06 ppm; ice 4.05/0.03 ppm), $O_3$ (137/159 ppm) and CO (60/43,54 ppb) above the CPT (0.21/0.82 km). (c) Evolution of ice and gas phase along the type A trajectory, derived from CLaMS-Ice and FDM models, initialized from in-situ measurements and compared with MLS observations of the gas phase near the trajectory. Time periods with available CALIPSO observations of ice are also indicated. Black squares denote the positions of the observations from which 60-day back- and forward trajectories were initiated.

of ice at the initialization time. The mean/maximum values for type B after 40 days are: 8.1/9.8 ppmv (CLaMS-Ice) and 6.5/7.9 ppmv (FDM). Only 14% (CLaMS-Ice) and 1% (FDM) of the initial observations did not experience any dehydration. The final positions of these non-dehydrated air parcels are within the LMS.

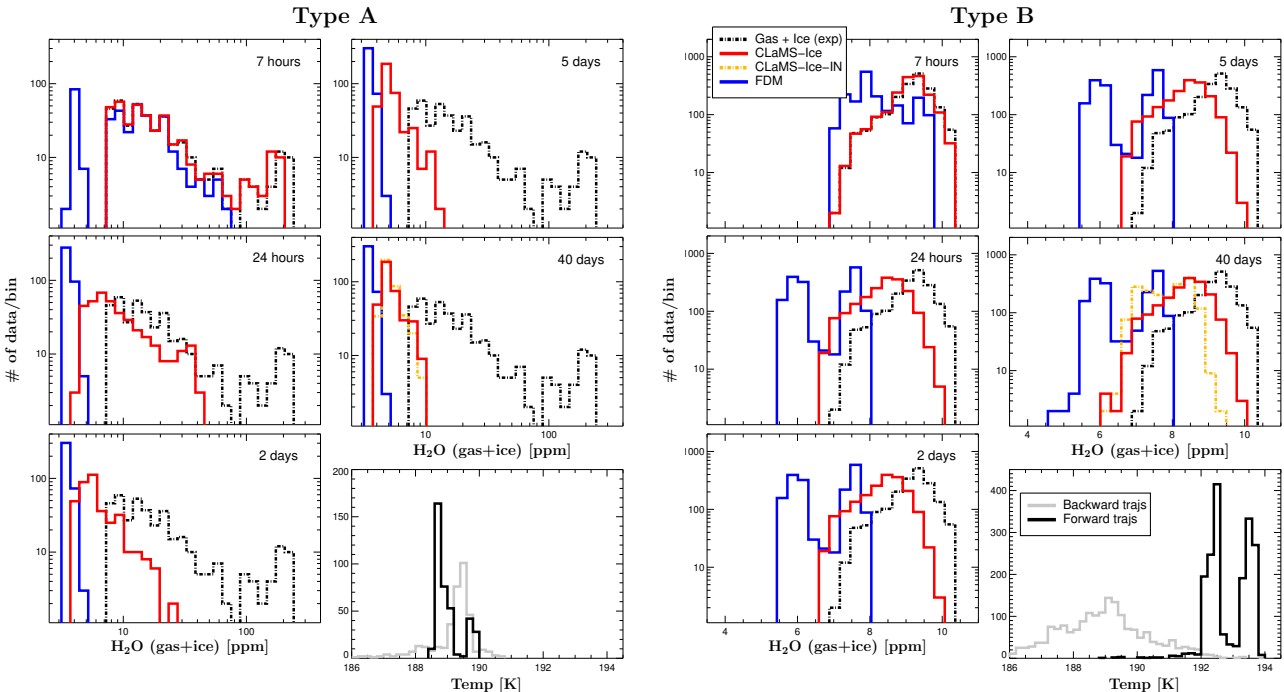

**Figure 3.** Time evolution of the total $H_2O$ frequency distribution (number of data points per bin) starting from two in situ observed distributions (left/right, type A/B, dashed black) as derived from forward trajectory calculations using CLaMS-Ice (red), FDM (blue), and from CLaMS-Ice with artificially enhanced heterogeneous ice nucleation (CLaMS-Ice-IN, orange); the latter is shown only 40 days after the initialization. For a better comparison, the initial frequency distribution is shown for all time steps (dashed black). Note that a logarithmic x-axis was applied for type A to take into account the large amount of ice used for initialization, while a simple linear x-axis was used for type B. The last panels for both type A and B air masses show the distribution of LDP temperature along the forward (black) and backward (grey) trajectories.

The dehydration scenarios for type A and B are consistent with the respective frequency distributions of LDP temperatures from back-and-forward trajectories (Figure 3, downward shifted panels). The strong dehydration of type A air masses, detrained very recently from fresh convection, is mainly due to the LDPs being experienced in the forward direction rather than along the backward trajectories. On the other hand, most of the air masses of type B, detrained from convection several days to weeks before, have already experienced their lowest temperatures in the past (cf. Khaykin et al. (2022)). But even for type B air masses, a significant dehydration can still be expected in the future, well above the CPT, at the southern edge of the anticyclone during the upward spiraling motion of the forward trajectories, as seen from the shift of the frequency distributions to lower mixing ratios on the right hand side of Figure 3.

## 4 Geographic perspective: Comparison with CALIPSO and MLS

We compare the ice distribution calculated by CLaMS-Ice during the dehydration periods along the forward trajectories with CALIPSO observations, which infer ice mixing ratios larger than ∼0.1 ppm based on the parameterization of in situ data (Avery et al., 2012). Figure 4 displays the results, where the horizontal (a) and vertical (b) large-scale temperature distributions are gray-coded and overlaid with the positions of the simulated and measured ice clouds, as well as the position of the ASM anticyclone (PV-based edge and the mean easterlies and westerlies) in August 2017. The comparison shows that type A generates significantly more ice than type B, and that type A agrees better with CALIPSO observations. Of the 442 type A ice events observed by CALIPSO, more than half are reproduced by CLaMS-Ice, while of the 132 CALIPSO ice events of type B, less than 6% are simulated by CLaMS-Ice, despite data set B being ∼ 5 times larger. The geographic positions of type A ice clouds are also better reproduced and are mainly found at 20N between 17 and 18 km altitude, with the strongest signature over North India. In contrast, type B ice simulated by CLaMS-Ice shows weaker signatures and a much larger horizontal spread, extending over the regions with coldest temperatures, mainly over southeast Asia and the Maritime Continent. These signatures seem to be related to isentropic mixing driven by Rossby waves, well-characterized by bent PV isolines surrounding the anticyclone (Konopka et al., 2009). There are a few weak CALIPSO signatures of ice in the LMS (type B) north of 35N between 400 and 420 K, which are not resolved by CLaMS-Ice. However, the expected warm temperatures in this part of the atmosphere raise some doubts about the origin of these signatures.

We also validate the calculated water vapor with the MLS observations along the forward trajectories. The comparison of CLaMS-Ice and FDM with MLS data for the type A observations is quite good. However, there is a significant disagreement for the type B data, even when ice nucleation is enhanced in the model (CLaMS-Ice-IN). We also observe a similarly strong disagreement when using FDM. A weaker disagreement was found for the type M data, consistent with its mixed properties (30% type A and 70% type B). However, including dehydration also along the backward trajectories (FDM-full), i.e. at the LDP being in the past, as done in many previous studies (Fueglistaler and Haynes, 2005; Liu et al., 2010; Smith et al., 2021), performs extremely well for all three data types: A, B and M, particularly in the region above 390 K (see also Appendix B).

## 5 Discussion and conclusions

The mere existence of moist plumes over the CPT, which are also well above the lapse rate tropopause and therefore in the stratosphere, does not necessarily imply persistent stratospheric moistening (Pan et al., 2019), as significant dehydration events along the forward trajectories are still possible, particularly in the regionally confined anticyclonic Asian monsoon circulation. In other words, a 1-D view is misleading, as observation of a moist air mass above the CPT at a particular time does not imply that the moisture remains in the stratosphere. One aspect is simply time variability, but more important is the fact that the Lagrangian time history determines dehydration, which requires consideration of the 4d temperature field (3d space + time), not just the temperature field in a single profile (1d space), or even the time-varying temperature field in a single profile (1d space + time).

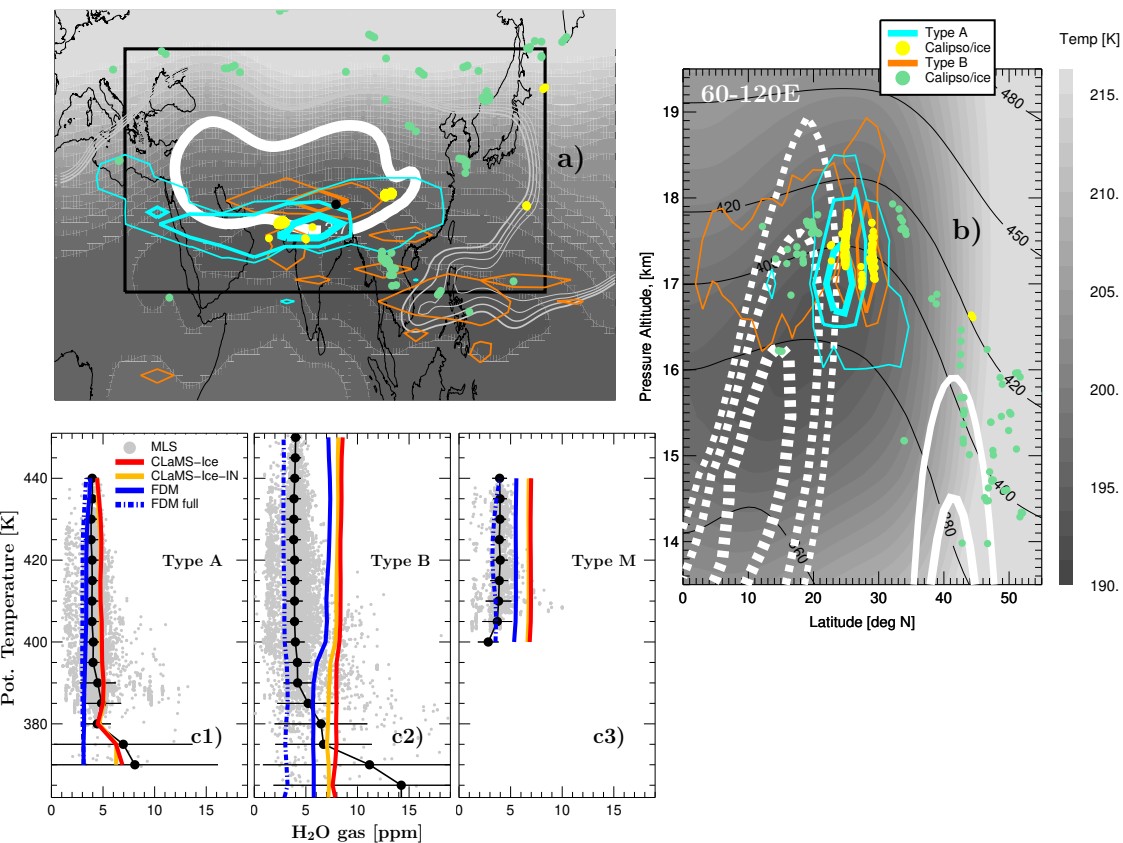

**Figure 4.** Geographic horizontal (a) and vertical (b) positions of ice formation (isolines of frequency distributions normalized by their total numbers) as derived from CLaMS-Ice applied along forward trajectories starting from the data sets A (cyan) and B (pink). The corresponding positions of CALIPSO ice observations in the vicinity of these trajectories are marked with yellow and green bold points. In the horizontal view (a), the edge of the ASM anticyclone (bold white) is derived from the gradient PV barrier at 380 K following the procedure described in Ploeger et al. (2017) (black filled circle - Kathmandu, the base of the StratoClim campaign). In the vertical view, the easterly and westerly jets (dashed and solid white lines) bound meridionally the ASM anticyclone. As the ice clouds resolved by CLaMS-Ice primarily result from the lowest temperatures, their geographic distribution is color-coded using a gray scale: (a) temperature minimum between potential temperature levels 360 and 420 K, (b) mean temperature averaged between 60-120 E, both from the ERA5 monthly mean for August 2017. The other PV isolines at 380 K (light gray lines between 5.8 and 6.2 PVU in (a)) indicate the position of the dynamical tropopause (Kunz et al., 2011). (c) Comparison with MLS for all trajectory parcels within the region confined by the black rectangle shown in (a) (gray – MLS data, black dots/horizontal lines – their mean values/standard deviations) split into data sets A, B, and M (c1-c3). Four models are used: CLaMS-Ice in the standard version (red), with the enhanced ice nuclei concentration (CLaMS-Ice-IN, orange) and by using a simple freeze drying model (FDM) along the forward and full trajectories (FDM/FDM-full, solid/dashed blue).

Based on our case study using StratoClim data, the importance of considering the full Lagrangian air mass pathway in both backward and forward direction is clear not only for type A air masses, where the absolute LDPs are still ahead, but also for type B air masses, where the absolute LDPs have already occurred a few weeks prior. Even for these cases, multiple and subsequent dehydration events at low temperatures, well below 195 K, can still occur during the upward spiraling in the ASM anticyclone. The highest ice concentrations are primarily located at the southern edge of the anticyclone, where the lowest ERA5 temperatures are observed (Figure 4). Our simulations with CLaMS-Ice reproduce CALIPSO ice observations well for type A air masses, but the agreement for type B is worse (Figure 4a and 4b). Note that the absolute LDPs of 75% of type A trajectories are in the future, with LDP ages ranging from 0-3 days. For type B trajectories, the LDPs have already occurred in the past, with ages ranging from -35 to -15 days. It is worth noting that our simplest approach, FDM, cannot reproduce any CALIPSO ice signatures, as ice is instantaneously formed and removed in this model setup.

To further evaluate the large-scale impact of the observed local hydration events, we compared different model results with MLS satellite observations. MLS has a significantly coarser vertical sampling resolution compared to CALIPSO, which can even resolve ice clouds extending over a few tens of meters. While the models exhibit good agreement with MLS observations for type A air masses, they consistently overestimate MLS, on average of 3-4 ppm for type B air masses when considering only forward trajectories (Figure 4c2). Despite the significant dehydration events along the forward trajectories, the excess water vapor observed in situ persists and is not captured by MLS. This wet bias relative to MLS cannot be eliminated even by applying CLaMS-Ice-IN, which artificially enhances dehydration to its highest realistic limits. Only when assuming dehydration at the LDPs along backward trajectories, typically occurring 2-3 weeks prior to the observations, do the results of FDM-full align well with MLS water vapor measurements. Therefore, the small-scale moist plumes observed above the CPT, likely originating from convective overshoots, are not relevant for explaining the large-scale water vapor budget.

This significant result prompts critical remarks. Firstly, moisture-rich type B air masses were independently observed by the FISH and FLASH instruments, with differences below 10%. Secondly, a potential warm temperature bias in the ERA5 data, estimated at approximately 1 K based on the differences to the observed temperatures along the flight track (Figure A1b), may account for a maximum of 0.5 ppm of the diagnosed difference (Fueglistaler et al., 2014). Other factors such as the quality of the trajectories, the matching criteria used to identify MLS observations, or the precise definition of the bounding rectangle (Figure 4a) are negligible (see Appendix B). Mixing may also play a role in smoothing out such moisture-rich structures. However, as no enhancement was diagnosed in the MLS observations, it would only support our statement that these structures are likely on small scales and not relevant for the large-scale water vapor budget. Finally, it is essential to note that our study is a single-case demonstration of the proposed mechanism's feasibility and seems particularly relevant for the Asian monsoon with its anticyclonic circulation regionally confining the ascending air masses. There are regions in the world, such as the American monsoon, where ice transport into the stratosphere could be more likely (Jensen et al., 2020; Park et al., 2021). Consequently, small-scale features may have a more substantial influence on the large-scale water vapor distribution in other regions. In such cases, additional case studies that follow our approach could provide further insights.

Despite these caveats, our results support some criticism related to the representativeness of local hydration events observed by in-situ measurements. E.g. this effect may influence the quantification of the SWV trends like those derived from the longest

available record of the balloon-borne, in situ NOAA frost point hygrometer over Boulder (Kunz et al., 2013; Hegglin et al., 2014; Lossow et al., 2017; Konopka et al., 2022). Our study shows that moist plumes can be sampled in the stratosphere which are not representative for the large-scale distributions of SWV. The fact that stratospheric satellite instruments capable of measuring SWV concentrations are approaching the end of their life time emphasizes the importance of setting up in-situ observation networks (e.g. using stratospheric balloons) with regular and "statistically robust" sampling (Müller et al., 2016).

Finally, a few remarks are necessary regarding the performance of our most idealized modeling approach, reconstructing the SWV from the absolute LDP derived from full back-and-forward trajectories covering several weeks. The trajectory-based reconstruction propagates the minimum saturation mixing ratio encountered at the LDP into the full 3D space. As the quality of temperatures around the tropopause has improved over the last decades, particularly for ERA5 (Tegtmeier et al., 2020), the quality of the reconstructed SWV has also improved. Our results show that tropopause temperatures exert a dominant control over the tropical stratosphere in the ASM region (Randel et al., 2015; Randel and Park, 2019), and this dominance seems to be more representative of global SWV values than sporadic observations of moist plumes in the stratosphere. However, we also found that the SWV reconstruction is not as effective for trajectories ending in the LMS (see Figure B1d). In this region, the final values of SWV are not only controlled by LDPs, but also by other processes such as mixing or downward transport of SWV affected by methane oxidation.

### Appendix A: ERA5-based trajectories, in-situ data and CLaMS-Ice

Both forward and backward 60-day trajectories used in this study start from the space-time coordinates of the in-situ observations collected on board the Geophysica aircraft. These trajectories are computed using the trajectory module of the Chemical Lagrangian Model of the Stratosphere (CLaMS) (McKenna et al., 2002), driven by the ERA5 horizontal wind velocities (Hersbach et al., 2020) and diabatic heating rates (Ploeger et al., 2010). The meteorological data used for the calculations have the highest available spatial resolution of $0.3 \times 0.3$ degree (137 model levels) and a temporal resolution of 1 hour (Hoffmann et al., 2019).

The merged back-and-forward trajectories, with a maximum duration of 120 days, are utilized to determine the Lagrangian dry point (LDP). Unlike in previous studies such as Ueyama et al. (2020) or Legras and Bucci (2020), the back-trajectories do not terminate at convective events. The vertical distance to the cold point tropopause (CPT) is defined as the geometric distance between the Geophysica flight track and the temperature minimum in the ERA5 temperature profiles, which are interpolated to the Geophysica flight track. The LDP is identified as the minimum saturation mixing ratio over ice, calculated from the ERA5 temperature and pressure data (Sonntag, 1994), interpolated along the forward and merged back-and-forward trajectories starting from the flight track.

The total water ($H_2O$) and ozone ($O_3$) measurements shown in Figure 1 were obtained using the Fast In situ Stratospheric Hygrometer (FISH) and the Fast-Response Chemiluminescent Airborne Ozone Analyzer (FOZAN), respectively. The FISH total water measurements inside ice clouds were corrected for inlet ice particle enhancements following the method described in Afchine et al. (2018), using the gas phase water measurements from the Fluorescent Lyman-Alpha Stratospheric Hygrometer

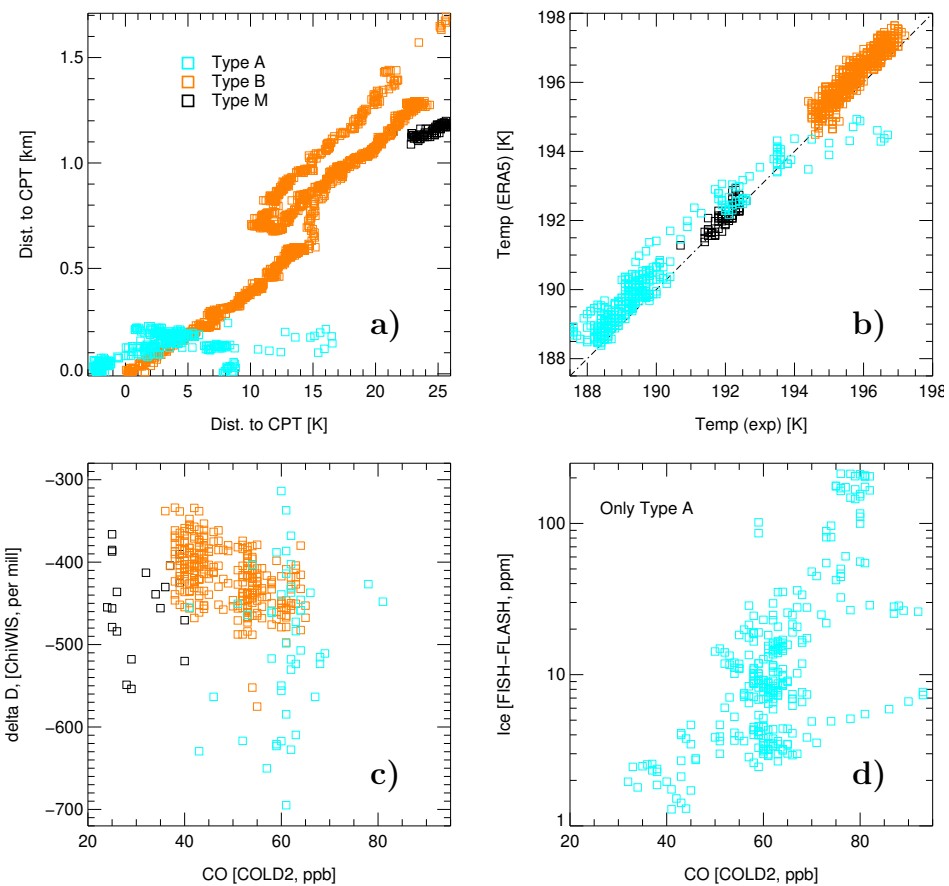

**Figure A1.** Additional properties of the type A, B, and M observations, color-coded as in Figure 1. (a) Distance to the cold point tropopause (CPT) in both geometric and potential temperature space. (b) Comparison between observed temperatures and ERA5 temperatures interpolated along the flight track of the Geophysica aircraft. (c) Correlation between CO and delta D values. (d) CO-ice correlations for type A data

(FLASH). For more detailed information on the FISH, FOZAN, and FLASH instruments, refer to Meyer et al. (2015) and Khaykin et al. (2022). Carbon monoxide (CO) concentrations were sampled using the Carbon Oxide Laser Detector 2 (COLD2) (Viciani et al., 2018).

190     Figure A1 provides additional details on the experimental data, specifically identifying the type A and B air masses as signatures of fresh and aged convection observed above the CPT, while type M represents mixed properties of type A and B. In Figure A1a, the geometric distance to the CPT (up to 1.7 km) is compared with the corresponding potential temperature difference (up to 26 K) derived from the ERA5 data. Particularly, type A air exhibits tightly packed isentropes ($\Delta\theta \approx 15K$ over $\Delta h \approx 0.3$ km), which is indicative of regions with strong convective activity.

Figure A1b evaluates the quality of the ERA5 temperature data by comparing them with the temperatures measured on board the Geophysica aircraft with the Thermo-Dynamic Complex instrument (Shur et al., 2006). The correlation between the two data sets is higher than 0.95, with a warm bias of ≈1 K in the ERA5 data, in agreement with Brunamonti et al. (2019). According to the Clausius-Clapeyron equation (Fueglistaler et al., 2014), a warm bias of ≈1 K can explain a difference of approximately 0.5 ppm between the simulated and observed water vapor mixing ratios.

In Figure A1c, the correlation between CO and Delta-D is shown, quantifying the isotopic ratios of water ($HDO/H_2O$) measured by the Chicago Water Isotope Spectrometer (ChiWIS). Delta-D values are enhanced (greater than -450‰) for water vapor molecules sublimated from convective ice clouds and depleted (less than -550‰) for data points representing the stratospheric background. Therefore, water vapor of type B (and partially type M) originates from convective ice clouds that have evaporated in the last 60-20 days. On the other hand, air masses of type A (and partially type M) exhibit signatures of fresh convection, with a wider spread of delta-D values, indicating that the transition from ice to the gas phase is only partially completed (Moyer et al., 1996; Sarkozy et al., 2020; Khaykin et al., 2022). Additionally, positive CO-ice correlations for type A data (Figure A1d) suggest fresh convection as a possible explanation.

CLaMS-Ice takes into account all relevant microphysical processes important for hydration and dehydration of air, such as nucleation of ice, diffusional growth, sublimation, and sedimentation processes that change the amount of water vapor and ice in the air parcel moving along the trajectory. The model, based on the two-moment scheme published by Spichtinger and Gierens (2009), has been extensively validated against measurements in cloud chamber experiments (Baumgartner et al., 2022). Although the ERA5 temperature interpolated along the trajectory is the main driver of all these processes, it can also be overlaid with temperature fluctuations induced by unresolved GW in the coarser meteorological fields, following the method described in Podglajen et al. (2016). CLaMS-Ice is initialized at the beginning of the forward trajectories with the in-situ observations, i.e. ice water content derived from the combination of FISH and FLASH instruments and ice particle number concentration in the range of 3-937 $\mu$m form the New Ice eXpEriment-Cloud and Aerosol Particle Spectrometer (NIXE-CAPS) (Krämer et al., 2016).

**Appendix B: Sensitivity studies (type B)**

For type B air masses, the modeled dehydration events in CLaMS-Ice or FDM do not exhibit sufficient strength to reproduce the significantly drier MLS observations observed along the forward trajectories. The wet bias present in our model simulations persists even when using CLaMS-Ice with GW parameterization and artificially enhanced ice nucleation rates. To further investigate this discrepancy, we conduct additional analyses to evaluate the robustness of our conclusion, which suggests that the disagreement is primarily attributed to the differences in representativeness between the highly resolved in-situ observations and the coarser MLS data. First, in Figure B1a, we illustrate the latitudinal dispersion of the type B trajectories across the Northern Hemisphere after three weeks of advection with the ERA5 winds. We compare these trajectories with those of type A and M. While the latter two remain confined within the ASM anticyclone, as indicated by the position of the jet (refer to Figure 4b), approximately 15% of the type B trajectories detach from the anticyclone and move into the lowermost stratosphere.

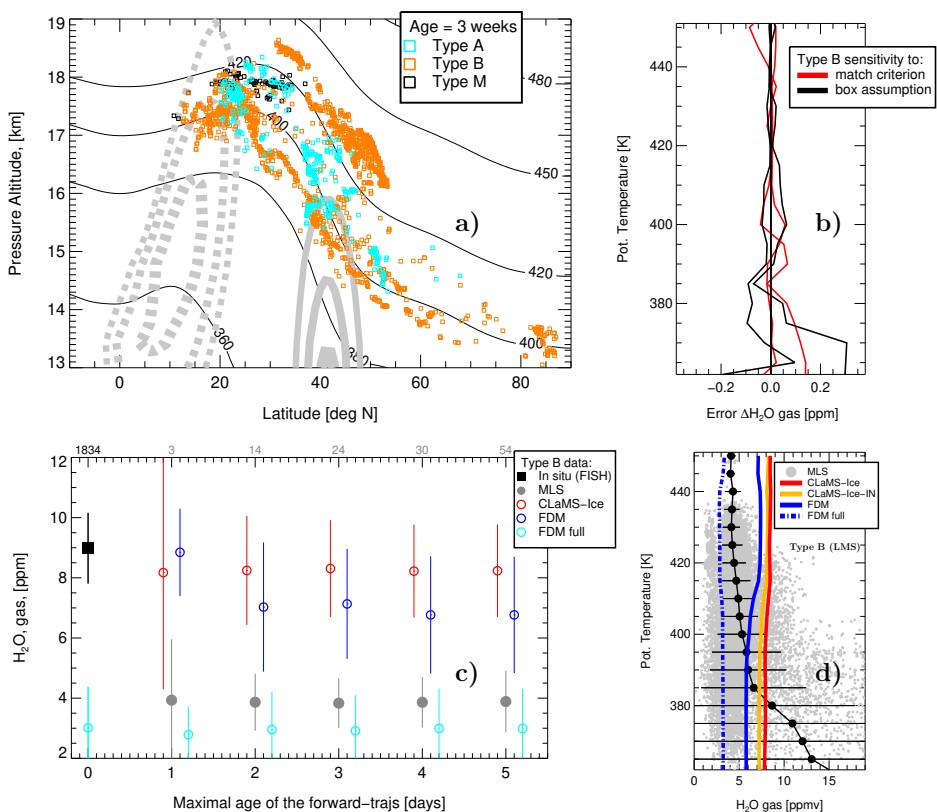

**Figure B1.** Sensitivity studies of dehydration scenarios for the type B data. (a) Lateral spread of trajectories after 3 weeks for all three data types (wind and isentropes as in Figure 4). (b) Sensitivity of the CLaMS-Ice profile shown in Figure 4c2 to the match criteria (strong, moderate, and weak) and the choice of the bounding rectangle of the ASM anticyclone shown in Figure 4a. (c) Sensitivity of the wet bias (relative to MLS) of the models to the maximum trajectory lengths considered (gray numbers indicate the number of available MLS observations). For a trajectory length of 0, the mean value over all 1834 type B observations is shown. For this case, the results of FDM-full are also available as they represent the simple minimum saturation mixing ratio of air at the LDP along the backward trajectories. Vertical lines represent the corresponding standard deviations. (d) Validation of four models for the subset of type B trajectories ending in the LMS.

To generate the mean profiles depicted in Figure 4c(1-3), we only include trajectories that remain within the bounding rectangle defined in Figure 4a. Additionally, three types of matching criteria, namely strong, moderate, and weak, were applied to identify the "nearest" MLS data points in terms of both time and space. Each matching criterion is characterized by different values for the distance in space and time between the trajectory position and the MLS overpass. Specifically, we used $\Delta t$ = 1h, 2h, 3h and $r$ = 100 km, 150 km, 200 km, with a vertical match criterion set at 20 hPa. The moderate version of the data match is considered as the default, as it represents a compromise between the number of matches and their quality.

In Figure B1b, we illustrate the weak sensitivity (less than 0.3 ppm) of the CLaMS-Ice mean profile, indicated by the red line in Figure 4c2, to the selection of match parameters and the precise bounding rectangle within a variability range of $\pm 5$ degrees latitude and $\pm 10$ degrees longitude. Likewise, a similar weak sensitivity was observed when comparing with CALIPSO data in Figure 4b. For this comparison, we utilized $\Delta t = 1$h, 2h, 4h and $r = 30$ km, 50 km, 150 km for the strong, moderate, and weak match criteria, respectively, with a vertical match criterion of 60 m.

In Figure B1c, we assess the sensitivity of the results presented in Figure 4c2 to the length of the forward trajectories. Generally, as the length of the trajectories increases, their quality decreases. Hence, we restrict our comparison of the model calculations (CLaMS-Ice, FDM) to trajectories spanning 1-5 days, initiated from the type B data points (1834 points in total). It should be noted that reducing the trajectory length leads to fewer encounters with MLS observations (indicated by the gray numbers). Throughout this analysis, we consistently compare the means and standard deviations of all successful matches, which highlight the wet bias in all model simulations compared to MLS, except for FDM-full (where full-backward and limited 1-5 days forward trajectories were employed).

Finally, in Figure B1d, we extend our analysis presented in Figure 4c1-c3 to include trajectories of type B that terminate in the LMS region instead of the bounding rectangle of the ASM anticyclone (as depicted in Figure B1a). It is important to highlight that this corresponds to less than 15% of the type B trajectories. The figure demonstrates that in this case, even FDM-full exhibits disagreement with MLS observations in the 380-420 K range. This discrepancy could be attributed to neglecting of processes such as mixing or downward transport of water vapor resulting from methane oxidation, or it may be due to reduced performance of MLS in this particular region of the atmosphere.

*Acknowledgements.* The authors would like to thank the European Centre for Medium-Range Weather Forecasts (ECMWF) for providing meteorological analysis for this study. We also appreciate the excellent programming support provided by Nicole Thomas. We thank Alexey Lykov, Alexey Ulanowski, and Vladimir Yushkov from the Central Aerological Observatory of Roshydromet, Dolgoprudny, Russian Federation, for their support related to the FLASH and FOZAN data. We are grateful to our Russian colleagues for their willingness to decline co-authorship. Funding for this work was provided by the CLaMS-ESM project of the Earth System Modelling Project (ESM), which also provided computing time on the ESM partition of the supercomputer JUWELS at the Jülich Supercomputing Centre (JSC). Jens-Uwe Grooß and Gebhard Günther made significant contributions to the data management on the supercomputer, and Bärbel Vogel assisted with trajectory calculations. This research has been supported by the Deutsche Forschungsgemeinschaft (DFG, German Research Foundation; TRR 301, project ID 428312742). Campaign planning and logistics were largely covered by the StratoClim project, funded by the European Commission's Seventh Framework Programme (FP7/2007-2013) under grant agreement no. 603557. We thank two anonymous reviewers for their valuable feedback and comments, which helped us to improve our manuscript. We extend our gratitude to Peter Haynes (editor of this paper) for moderating the dialog with the reviewers. Finally, we would like to thank chatGPT (https://chat.openai.com, last accessed: 10 March 2023) for their assistance in improving the final text.

*Code and data availability.* The trajectrory module is a part of CLaMS-2.0/MESSy code based on MESSy version 2.54 and accessible for MESSy consortium members at https://gitlab.dkrz.de/MESSy. The usage of MESSy and access to the source code is licensed to all affiliates of institutions which are members of the MESSy Consortium. Institutions can become a member of the MESSy Consortium by signing the MESSy Memorandum of Understanding. More information can be found on the MESSy Consortium Website (http://www.messy-interface.org, last access: 30 June 2022). ERA5 model level reanalysis data are available from the ECMWF as deter-

ministic forecasts (atmospheric model): via: https://apps.ecmwf.int/data-catalogues/era5/?class=ea. The StratoClim data can be downloaded from: http://www.stratoclim.org/. For more detailed model data, please contact the authors.

*Competing interests.* Marc von Hobe and Martina Krämer are members of the editorial board of Atmospheric Chemistry and Physics. All authors have no competing interests.

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
