# Peer review of "The dehydration carousel of stratospheric water vapor in the Asian Summer Monsoon Anticyclone"

_EGUsphere, 2023_

## Author Comment (AC1)

Response to reviewer 1

We would like to thank reviewer 1 for a very thoughtful and detailed review of our manuscript that has greatly contributed to the improvement of the paper. To address the recommendations of both reviewers regarding the readability of the graphics and the length of the captions, we have made the following changes:

1. Figures 1 and 2 have been simplified and cleaned up.

2. Figures 3 and 4 have been slightly enlarged, and redundant notation has been removed.

3. Both figures in the appendix have been revised to incorporate the additional requests made by the reviewers (see below)

4. The captions have been significantly shortened. Relevant portions have been moved to the main text or the appendix.

5. Most of the changes to the text have been made in the appendix. We would like to note that we are still striving to meet the requirements of ACP Letters, which include a limit of 2500 words for the main text and 200 words for the abstract.

6. The title has been changed to: "The dehydration carousel of stratospheric water vapor in the Asian Summer Monsoon Anticyclone"

In the following, we address all the points (marked in blue) that were raised in the review (denoted by italic letters).
Major comments:

1. *This is an important paper, and it provides good evidence that "convective moistening of the stratosphere" over monsoons is far more complicated than some of the earlier proponents have envisioned. Previous discussions of the convective moistening process have assumed that Convection reaching above the tropopause simply resets the relative humidity to saturation. In this study, it is evident that the air parcels may continue to dehydrate due to the elevated cold points as they move around in the Asian monsoon anticyclone. In my own mind, the higher the convection, the colder the air due to adiabatic expansion and the more complete the dehydration. This paper shows that parcels launched at the top of other convective events, can transit through colder air undergoing later dehydration.*

   Thanks a lot for these nice words

2. *I found the abstract quite descriptive and useful. I recommend that a longer version of the abstract be repeated in the summary section which could be expanded.*

   Thank you for your positive feedback on the abstract. We understand your suggestion to include a longer version of the abstract in the summary section. However, we are constrained by the word limit imposed by ACP Letters, which restricts us to 2500 words in the main text. Nonetheless, we have made some efforts to provide a comprehensive overview of the study within the given constraints.

3. *In general, the figures are hard to read, and the captions are too long. The author might consider breaking up the figures into smaller groups and edit the captions. I would delete Fig. 5, not very helpful.*

   See our explanation above. We understand your point regarding Fig. 5. While it may seem trivial to those familiar with the topic, it serves an important purpose in addressing a common misconception prevalent in the experimental community. There is a widespread opinion that moistening above the cold point tropopause (CPT) can be considered as "irreversible" moistening of the stratosphere, completely neglecting the Lagrangian view. Given this context, I believe it is valuable to include Fig. 5 in the manuscript. In fact, it played a crucial role in motivating the writing of this paper.

Minor comments:

1. *I would reference Brewer (1949) as the originator of the CP regulation of water vapor theory.*

   was done

2. *The introduction is too brief to cover this complex and important scientific field. For example, you might also expand on some of the previous publications mentioned. The Randel & Park (2019) paper is a particularly important prelude to these conclusions. Additionally, there are additional trajectory model simulations by Ueyama & Schoeberl and collaborators are relevant - these papers also used convection and ice formation models. The Avery paper focusses on El Nino, not the monsoon. Lumping the regular monsoon convection system with El Nino seems like a stretch to me.*

   We appreciate the reviewer's feedback. The introduction has been expanded accordingly. The citation for Ueyama et al. (2023) has been included, and the publications have been categorized into regular convection and monsoon convection for better clarity.

3. *Table 1 is confusing. 10.08 flights? Is this the date? Why is this relevant? I would put a comment in the Table on the difference between Type A and Type B. Perhaps a comment line "recent convective influence" and "aged convective influence" for A and B - something that the reader can immediately grasp*

   great recommendation. we followed this idea and changed the table and the text around it

4. *I would put the references to the instruments in Figure 1 caption into the text. All the references make the caption difficult to read. "time distance?" you mean time since encountering an LDP.*

   The references were moved to the appendix. and the formulation "time distance" was replaced by the "LDP age" and explained both in the main text and in the caption of Fig. 1

5. *The exact LDP is a little uncertain since gravity waves could create an LDP even after the temperature along the path has warmed up a little. I assume you observed temperature fluctuation measurements as part of the aircraft flights. You could translate this into an uncertainty in the LDP time using Delta-T and the temperature along the path. These fluctuations could be important. It wasn't clear from the text that Podglajen et al. (2016)*

*gravity wave parameterization is included, or if it is included, does it match observations over mountainous Himalayas?*

Yes, we agree that the exact definition of the LDP is a little uncertain since gravity waves may influence it. Our definition is based solely on the temperature fields resolved in ERA5 data. However, CLaMS-Ice applied in this paper uses the Podglajen et al. (2016) parametrization of gravitiy waves, which follows a statisical approach to represent the missing temperature fluctuations in ERA5. Note, as we discuss it (caption of Fig. 3), the influence on our results is small.

6. *You might add some additional references on CO photolysis beyond von Hobe (2021). CO is measured by MLS. Minschwaner et al., (2010) is the classic paper on CO lifetime, also see Liang et al. (2023) and references therein.*

   Minschwaner et al., 2010 reference is now included.

7. *Clearly type B is "aged air" with higher ozone, lower CO whereas type A is "younger air". So it was a little surprising to see the LDP age for type A all over the map (Fig.1 C). This confusing point was straightened out in Fig. 1d so maybe 1c could be eliminated or make the symbols smaller.*

   great recommendation, we removed figure 1c and simplified in this way our story following general recommendations of both reviewers

8. *FIG. 2 - it might be useful to locate where the Part b Lagrangian dry point is located on the map shown in Part a. I would have shown the type A trajectory in 2c - makes your point better - and put the Type A label inside 2d. Remove the not-needed information from caption of Fig. 2*

   we included now two panels in Fig 2 with the full age of air derived from the back-and-forwar trajectories

9. *Line 80. CALIPSO does not detect ice mixing ratios. It detects particles and then using a model the ice mixing ratios are inferred... maybe "..which can be used to infer ice mixing ratios (Avery et al., 2012)".*

   was included, see L86: "We compare the ice distribution calculated by CLaMS-Ice during the dehydration periods along the forward trajectories with CALIPSO observations, which detect ice mixing ratios larger than ∼0.1 ppm (Avery et al., 2012)."

10. *Fig. 4 caption, although way too long, was actually readable.*

    ...one sentence was removed, a new citation was added following the recommendation of the other reviewer

11. *How does the aircraft temperatures compare with ERA5. The type B trajectories will encounter ERA5 temperatures, if these temperatures are too warm and you are downstream from the coldest temperature, then you might see a bias. Can you validate these temperatures against GNSS-RO?*

The comparison between the in-situ temperature measured onboard of the Geophysica with the ERA5 data interpolated at these points is now discussed in Fig. A1b. We did not includ any other validation.

12. *How do you account for the vertical averaging kernel in the MLS measurements?*

   We do not account for the MLS vertical averaging kernel. We are aware in the huge differences in the vertical and horizontal resolution between MLS (3 km/200 km, respectively) and the spatialy higly resolved in-situ observations. The missing agreement between these two data sets, especially the wet bias of the in-situ observations is interpreted in this study as a non-repesentativeness of our in-situ observations for the large-scale distributions sampled by MLS. We discuss now this point in much more details in Fig B1b and B1c, see also the related text.

13. *Line 100 Schoeberl and Dessler used forward trajectories.*

   was corrected

14. *I think some explanation on what is done with full trajectories is needed. Does the full start at the measurement point and go backward X days, or - like Ueyama et al. (2023) does it terminate at convection?*

   Thanks for this remark. We use the full back-and-forward, in total 120 days trajectory to determine the LDP and do not terminate the backward trajectories above the convective towers. This explanation is now included in L164-5

15. *Line 111 "highest ice concentration found mainly at southern edge." Where the temperatures are coldest according to Fig. 4.. might want to point that out.*

   good idea...the respective sentence was completed

16. *Line 117 ...vertical sampling resolution than CALIPSO*

   The word "vertical" is now included

17. *Line 124 "are not able to freeze out the excess water" ...assuming the temperatures from ERA5 are correct and there are no gravity waves. How much colder would the temperatures have to be to get the right water vapor? I suspect only a couple degrees...*

   We compare now temperatures observed onboard of the Geophysica with the ERA5 temperatures. See Fig. 1Bb and the related text.

18. *Line 126.. I am confused about the backward trajectories. Presumably you start with the aircraft measurement of water and you go backward in time to get a temperature field.*

   Our approach is indeed simpler. We utilize the Lagrangian dry point of the backward trajectory to calculate the minimum saturation mixing ratio, which we refer to as FDM-full. This information is described in the caption of Figure B1c in the revised form of the manuscript.

19. *Then starting with a saturated parcel at the furthest back time where it has encountered convection, you dehydrate and arrive at the predicted measurement. Do the two values of water agree? I am wondering if the instrument measured air might be wet biased. Do they*

*agree with MLS? I think that this weird Type B bias needs more discussion as to possible sources of error.*

Based on the reviewer's ideas and questions, we have included a new Figure B1c to address these points. Our findings consistently demonstrate that regardless of how we approach the analysis, in situ data of type B are consistently nearly twice as large as MLS observations. While there is still potential for factors such as "missing microphysics," gravity waves, trajectory errors, or match errors to contribute to this discrepancy, we believe that the non-representativeness of these small-scale structures appears to be the most probable explanation.

20. *I would delete Fig. 5. I found it confusing and not helpful.*

In response to the reviewer's comment, we understand the confusion and concerns regarding Figure 5. However, we believe that including this figure is important for pedagogical reasons. One of the key motivations behind our paper was to address the misconception, often held by experimental researchers, that being above the Cold Point Tropopause (CPT) automatically implies being in the stratosphere. The figure helps to illustrate and clarify that rather a Lagrangian than an Eulerian view matters, especially in relation to water vapor. We are open to suggestions on how we can improve the figure to make it clearer and more useful for the readers.

---

## Author Comment (AC2)

Response to reviewer 2

We would like to thank reviewer 2 for a very thoughtful and detailed review of our manuscript that has greatly contributed to the improvement of the paper.helped to improve the paper. To address the recommendations of both reviewers regarding the readability of the graphics and the length of the captions, we have made the following changes:

1. Figures 1 and 2 have been simplified and cleaned up.

2. Figures 3 and 4 have been slightly enlarged, and redundant notation has been removed.

3. Both figures in the appendix have been revised to incorporate the additional requests made by the reviewers (see below)

4. The captions have been significantly shortened. Relevant portions have been moved to the main text or the appendix.

5. Most of the changes to the text have been made in the appendix. We would like to note that we are still striving to meet the requirements of ACP Letters, which include a limit of 2500 words for the main text and 200 words for the abstract.

6. The title has been changed to: "The dehydration carousel of stratospheric water vapor in the Asian Summer Monsoon Anticyclone"

In the following, we address all the points (marked in blue) that were raised in the review (denoted by italic letters).
Major comments:

1. *Title/Abstract*

   *In the title and abstract, the discussion of high-concentration water vapor content appears, but the main text is more about the relationship between the formation of low-concentration water vapor content above the CPT and ice clouds. It would be better to change the title/abstract along the main subject or vice versa. If you want to investigate the behavior of high-concentration water vapor above the CPT, why not analyze cases where water vapor is more than 7ppmv shown in Fig.1a? It is also not clear the reason why Type B and M were used for analysis. Please describe the reason to select those cases.*

   We initially started with all the local flight data and then applied specific conditions, including a water vapor concentration larger than 7 ppm and ozone concentration larger than 100 ppb, while focusing on data observed above the cold point tropopause (CPT). This selection process resulted in three distinct types of data: A, B, and M. Type A data primarily represents air with signatures of fresh convection, while type B data consists of air with aged convection, and type M data represents mixed properties. All these 2315 data points are relatively moist (>7 ppm) and in the stratopshere (above CPT with $O_3 > 100$ ppm.
   Our main focus in the paper is to demonstrate the potential for significant dehydration effects along the forward trajectories, despite the initially high water vapor concentrations observed above the CPT. We emphasize the importance of the "Lagrangian view," which highlights

the possibility of moisture depletion along these trajectories. This challenges the common assumption of strong and irreversible moistening of the stratosphere based solely on the "Eulerian view." We have also changed the title of our paper.

2. *Structure of manuscript*

   *Many explanations and interpretations (and it is important!) of the figures were written on the appendix, and it was very difficult to read. Implying that it may be a difficult task to summarize in a short report. For example, "fresh convection signature" in Figure 1 was explained in the caption of Figure A1 of Appendix A, and the explanation of the trajectory analysis and the definition of CPT were in Appendix A those are important matter for this study.*

   Sorry for that. We included now some changes of figures, captions and text to improve the readability of the paper (see above). Both, definition of the CPT and LDP are in the main text.

3. *Representativeness of selected data*

   *Regarding the effect on water vapor in the lower stratosphere, which is the purpose, there is a gap because the trajectory analysis is performed with only a limited number of observations (Type A,B and M). Further, related with my major comment (1), it is unclear whether the selected cases are actually representative ones. It is recommended that you add the reason for the selection and add an evaluation of its validity.*

   Of course, the avaiable data can be only interpreted as a case study. We state this point very clearly in the "Discussion and conclusions" part. See L132-141

4. *Quality of trajectory model*

   *The trajectory model used for the analysis may be the latest, but what is the guarantee of the reliability of the expression of supersaturation and vertical flow (here, diabatic heating is substituted), which exist in many cases? Shouldn't those differences (between observation and simulation) also be included in the discussion section?*

   Yes we agree. In the revised version of the manuscript, we included a section related to the quality of the ERA5 temperatures (Fig. A1b) and to the dependence of our results on the length of the trajectories (Fig. B1c) as the quality of the trajectory decreases with its lengths.

Minor comments:

1. *Line numbers in odd pare are missing from page 3.*

   was corrected

2. *p.2, l.26:*
   *How scales of "the large-scale moisture budget" for temporal and horizontal? How scales of "the large-scale moisture budget" for temporal and horizontal?*

This is probably difficult to derive from only a few statistically non-representative in situ observations. However, a well-validated model could provide insights into the scaling properties of the large-scale moisture budget in terms of temporal and horizontal scales. This could be an area of investigation for future studies.

3. *p.2, l.28:*
   *the subtitle should be changed, for example in situ and behavior of selected data etc..*

   was changed

4. *p.2, l.30:*
   *need the explanation of definition of "the local CPT" here and the "local" means unclear.*

   was done

5. *p.3, l.1:*
   *as a function of "vertical" distance to the CPT. Let me confirm that this distance was defined in either direction (up/down or above/below).*

   good point!, was done

6. *p.3, l.4:*
   *Figure 1c: "fresh convection signatures" In the case of "fresh", the CO concentration is likely to be high due to the air from troposphere, but this is inconsistent with the fact that the CO concentration has a range from 30 to 100 ppb. How can this be explained? In relation to this, does the distribution of LDP age from 60 days before in Fig1c mean that it has been present in the lower stratosphere since 60 days before? If so, can you say "fresh"? This reviewer confuses this point.*

   Sorry for the confusion. Following the recommendation of another reviewer, we have revised the paper to consistently classify data types A and B as signatures of fresh and aged convection, respectively, while data type M represents a mixture of these properties. It is important to note that these classifications are more statistical in nature. For example, data set A may include air masses with properties indicative of aged convection, such as Lagrangian Dry Point (LDP) values from 3 weeks in the past. This variability may be attributed to errors in the ERA5 data, which generally reproduces the observed properties of data set A but may not for every individual data point within this data set.

7. *p.3, Fig1:*
   *To understand the positional relationship in the vertical direction, it is useful to have the potential temperature distribution.*

   good point, was done, see Fig. A1a

8. *p.4, l.48:*
   *Considering supersaturation, should it be removed?*

   The formulation was slightly changed

9. *p.4, l.63-64:*
   *"where..." Is it not fully represented by FDM, or does it look like it has not been removed because the fluctuation of FDM is small?*

GW are not included in FDM and this is not the reason for the differences to CLaMS-Ice. Much more, in FDM, ice is removed insantanously while in CLaMS-Ice ice removal occurs through the interplay of condensation, evaporation, and sedimentation. The respective text was reformulated

10. *p.4, last paragraph:*
*On Type A, the temperatures are rising in the first three days (Fig 2c), but there is a lot of ice (Fig2d). In Fig 3, is it due to the sedimentation of the ice that the decrease in water vapor is remarkable after 7 hours and 2 days?*

At the beginning, the temperature is around 192K with some small-scale fluctuations. During the first two days, the temperature remains relatively stable, and sedimentation becomes the dominant process that significantly affects the water vapor distribution. By day 3, the temperature increases to 195K. During this period, sublimation of ice crystals becomes increasingly important but does not significantly influence the distribution. This is why the distribution between 2 and 5 days does not change dramatically.

11. *p.5, bottom of Fig 2c:*
*It is better that the y-axis direction is opposite. It is easy to understand to express that the temperature is low at the top, that is, the altitude is high.*

To simplify our story and to reduce the number of figures, this panel was completely removed as Fig 2b explains most of this behaviour

12. *p.5, l.2 in caption of Fig 2.:*
*in Figure 1 (a). Three representative... (need comma)*

Comma was included

13. *p.5, l.5 in caption of Fig 2.:*
*replace to slash, 60/43/54*

was done

14. *p.6, last sentence of caption in Fig3.:*
*back (grey) and forward (black)*

was done

15. *p.7, l.9 in caption of Fig 4:*
*light grey -> white? In general, the 2-3PV is defined as dynamical tropopause. The value of 5.8-6.2 PVU is higher than general.*

As ashown in Kunz et al. JGR, 2011, at lower $\theta$-levels around 350 K, dynamical tropopopause around 2-3 PVU is a good definition. However, at $\theta$=380 K, like in this figure, higher values between 5 and 7 PVU are more appropriate. This was the motivation for our choice. The respective citation is now included

16. *p.8, l.88:*
*on "spread" for Type B; Can't you visualize how much the trajectory is "spread"? Is it possible to add an example of spreading to Fig2a?*

was done, see Fig. B1a

17. *p.8, l.112:*

    *Is it correct (good correlation) to focus only on the south side of ASM in the case of Type B?*

    Now we discuss in Fig B1a the spread of the type B air, in particular the part detached from the ASM anticylone and transported into the lowermost stratosphere. This part is also analysed in Fig B1d and the related text

18. *p.9, l.121:*

    *Figure 5(c) -> Figure 4(c)*

    was done

19. *p.9, Fig5:*

    *The horizontal and vertical lines for supplementation in the figure are too thin to be seen.*

    the lines are now thicker

---

## Referee Report (RR1)

2nd review of "The dehydration carousel of stratospheric water vapor in the Asian Summer Monsoon Anticyclone" submitted by Konopuka et al. to EGUsphere.

Thank you for replying the reviewer's comments and questions. Almost replies were satisfied, however the following point is critical and the authors re-consider the point.

Major comments:

It is needed that the authors to consider whether Fig.5 is needed or not, because it is likely to mislead the reader, as it is actually the comprehensive interpretation drawing from the different air masses (both in time and origin). This reviewer recommends either remove Fig. 5 or left the Lagrange diagram (remove Euler part, left of Fig5.) with time-altitude-location information (by figure or description in text) clearly.

---

## Referee Report (RR2)

Review of Revised Version of "The dehydration carousel of stratospheric water vapor in the Asian Summer Monsoon Anticyclone"

This paper is significantly improved and important. There are still a few minor issues as indicated below.

Fig. 5 is more confusing than helpful. You make your main carousel point in Fig. 1, the 1D representation is not helping since the LDPs in 5b are widely separated in space. I suggest you drop Fig.5.

I am still a little puzzled why Type B water vapor (Fig. 4) is so much higher than A or M in the forward trajectory case. The explanation given in lines 125 to 141 suggests the following. Looking at Fig. 2b, the type B trajectory was dehydrated maybe 12 days earlier than sampled and since no further dehydration takes place after sampling the water vapor theoretically remains at LDP values – however the water vapor amount is larger than MLS. In the "full Lagrangian" you start 60 days earlier and run the trajectory 120 days forward. Then you get good agreement. This means that the measured water (the start of the B forward trajectory) is too high. There are two possibilities (1) the instrument is biased relative to MLS (2) some mixing took place with wetter air before it got to where it was sampled. In any event, perhaps the authors could expand on this dilemma somewhat. I found the discussion 133-142 inadequate.

Minor comments.

ln 10 ... LDP mixing ratios
ln 14 ... add Dessler et all. (2013) [Dessler, A.E., M. R. Schoeberl, T. Wang, S. Davis, K. H. Rosenlof, (2013) Stratospheric water vapor feedback, PNAS, www.pnas.org/cgi/doi/10.1073/pnas.1310344110]

Ln 17  In particular.. This sentence is awkwardly worded and somewhat confusing.
Ln 24 "for the large-scale SWV distribution" how about  "in order to reconstruct the large-scale.."

Fig. 1 caption  ... (see text and Table 1 for details)

Fig. 2 I presume that the square is the starting point of the back and forward trajectories. Please say that in the caption.  Shouldn't there be two squares on Fig. 2a? Or are they on top of each other?

ln 87  'which implies ice mixing ratios larger that ~0.1 ppm'
As I noted in the previous review, the CALIPSO doesn't detect ice mixing ratios. The mixing ratio is parameterization based on Heymsfield *in situ* data and is inferred.

Fig. 4 caption "As the ice clouds resolved by CLaMS-Ice are mainly  produced by the lowest temperatures.

Ln 135  Fueglistaler et al. (2014) used ERAi data. Are the biases the same?

Ln 143 in situ

---

## Author Response (AR2)

Response to the remarks of both reviewers and to editor's comments

In the revised version, the schematic Fig. 5 was removed. This change was made in response to the main criticism of both reviewers. We also included some new text, primarily following the recommendations of Peter Haynes (editor) (see lines L111-L118 in the revised version). We believe that by taking this approach, our point is now formulated more clearly, and we avoid any potential confusion that might arise from a simplified schematic figure. We extend our gratitude to Peter Haynes for moderating this issue. There are still a few minor comments from reviewer 1, which are addressed below.

Minor comments:

1. *I am still a little puzzled why Type B water vapor (Fig. 4) is so much higher than A or M in the forward trajectory case. The explanation given in lines 125 to 141 suggests the following. Looking at Fig. 2b, the type B trajectory was dehydrated maybe 12 days earlier than sampled and since no further dehydration takes place after sampling the water vapor theoretically remains at LDP value – however the water vapor amount is larger than MLS. In the "full Lagrangian" you start 60 days earlier and run the trajectory 120 days forward. Then you get good agreement. This means that the measured water (the start of the B forward trajectory) is too high. There are two possibilities (1) the instrument is biased relative to MLS (2) some mixing took place with wetter air before it got to where it was sampled. In any event, perhaps the authors could expand on this dilemma somewhat. I found the discussion 133-142 inadequate.*

   Yes, we agree that the results concerning type B are indeed surprising. Although also potential errors in the instruments or reanalysis-based simulations could play a role, we think that the most likely explanation for the high-bias of type B trajectory water vapor are moist small-scale features. These small-scale features are observed by the in-situ instrument but not by MLS. As they appear unimportant to explain the large-scale distribution as observed by MLS, we conclude that these features are not relevant for the large-scale water vapor budget. Mixing may also play a role in smoothing out such moisture-rich structures. However, as no enhancement was detected in the MLS observations, it only strengthens our statement that these structures are likely on a small scale and not relevant for the large-scale water vapor budget. In response to the reviewer's criticism, we have slightly revised the paragraphs between L129 and L151 The question of why the "full Lagrangian" reconstruction based on the LDP 2-3 weeks before the observation performs so well is discussed in the final paragraph of the "Discussion and conclusions" section.

2. *In L10 ...LDP mixing ratios*

   was done

3. *In L14 add Dessler et all. (2013)*

   was done

4. *In L17 In particular.. This sentence is awkwardly worded and somewhat confusing.*

   This sentence was reformulated, see L16-20

5. *In L24 "for the large-scale SWV distribution" how about "in order to reconstruct the large-scale.."*

   was replaced

6. *Fig. 1 caption ...(see text and Table 1 for details)*

   was included

7. *Fig. 2, I presume that the square is the starting point of the back and forward trajectories. Please say that in the caption. Shouldn't there be two squares on Fig. 2a? Or are they on top of each other?*

   additional sentence was included. It is correct that both squares in Fig. 2a are on top of each other.

8. *L87 "which implies ice mixing ratios larger that 0.1 ppm", As I noted in the previous review, the CALIPSO doesn't detect ice mixing ratios. The mixing ratio is parameterization based on Heymsfield in situ data and is inferred.*

   The sentence was reformulated following the recommendation

9. *Fig. 4 caption "As the ice clouds resolved by CLaMS-Ice are mainly driven by the experienced (produced) by the lowest temperatures."*

   The sentence was reformulated, see L88-90

10. L135 Fueglistaler et al. (2014) used ERAi data. Are the biases the same?

    We use the citation Fueglistaler et al. (2014) to apply his scaling factor between the CPT and the corresponding water vapor. In Fueglistaler et al. (2014) there is no comparison with in-situ data and, consequently, ERA-I bias was not calculated.

11. L143 in situ

    was done